# Relationship between true digestibility of dietary phosphorus and gastrointestinal bacteria of goats

**Lizhi Wang**[1,2][*], **Ali Mujtaba Shah**[1,2,3‡], **Yuehui Liu**[1,2], **Lei Jin**[1,2‡], **Zhisheng Wang**[1,2], **Bai Xue**[1,2], **Quanhui Peng**[1,2]

**1** Institute of Animal Nutrition, Sichuan Agricultural University, Chengdu, Sichuan, China, **2** Key Laboratory of Animal Disease-Resistant Nutrition, Chengdu, Sichuan, China, **3** Department of Livestock Production, Shaheed Benazir Bhutto University of Veterinary and Animal Science, Sakrand, Sindh, Pakistan

☯ These authors contributed equally to this work.
‡ These authors also contributed equally to this work.
* wanglizhi08@aliyun.com

**Data Availability Statement:** All the data of sequence in current research were placed in NCBI's sequence read archive (SRA) database with the accession No. SRP185613.

## Abstract

The present research was conducted to evaluate the connection between the true digestibility of Phosphorus (TDP) in diet and bacterial community structure in the gastrointestinal tract (GIT) of goats. Twenty-eight Nubian goats were chosen and metabolic experiment was conducted to analyze TDP of research animals. Eight goats were grouped into the high digestibility of phosphorus (HP) phenotype, and another 8 were grouped into the low digestibility of phosphorus (LP) phenotype. And from the rumen, abomasum, jejunim, cecum and colon content of the goats, bacterial 16S rRNA gene amplicons were sequenced. In the rumen 239 genera belonging to 23 phyla, in abomasum 319 genera belonging to 30 phyla, in jejunum 248 genera belonging to 36 phyla, in colon 248 genera belonging to 25 phyla and in cecum 246 genera belonging to 23 phyla were noticed. In addition, there was a significant correlation between the TDP and the abundance of *Ruminococcaceae_UCG-010*, *Ruminococcus_2*, *Ruminococcaceae_UCG-014*, *Selenomonas_1* and *Prevotella* in the rumen, *Lachnospiraceae_ND3007_group*, *Saccharofermentans*, *Ruminococcus_1*, *Ruminococcaceae_UCG-014*, *Lachnospiraceae_XPB1014_group* and *Desulfovibrio* in the abomasum, *Prevotella*, *Clostridium_sensu_stricto_1*, *Fibrobacter*, *Desulfovibrio* and *Ruminococcus_2* in the jejunum, *Ruminococcaceae_UCG-014* in the colon, and *Desulfovibrio* in the cecum. Present research trial recommended that the community of gastrointestinal microbiota is a factor affecting TDP in goats.

## Introduction

Phosphorus is a major mineral element that affects the growth and development of animals and acts a vital biological part in the formation and growth of skeletons. Young animals develop rickets, and adult animals present osteomalacia when animals lack phosphorus. In addition, phosphorus also participates in the digestion and metabolism of almost all nutrients,

**Funding:** This study was funded by the National Key R&D Program of China (grant number: 2017YFD0502005) and Sichuan Beef Cattle Innovation Group (grant number: 035Z389).

**Competing interests:** The authors have declared that no competing interests exist.

such as amino acids, lipids and carbohydrates, in animals in the form of nucleic acids, phosphoproteins and phospholipids [1]. It was found that phosphorus in plant feedstuff is poorly or not at all utilized by monogastric animals due to phytic acid (myo-inositol hexaphosphoric acid) binding. The undigested phytate phosphorus is excreted with feces, which causes severe environmental pollution problems, contributing to surface water eutrophication in locations where there is intensive monogastric livestock production [2, 3]. Phytase is an enzyme that catalyzes the hydrolysis of phytic acid found in grains and oil seeds to myo-inositol and inorganic phosphate, which are then absorbed in the small intestine [4]. Many previous experiments have proven that supplementation of monogastric diets with phytase not only significantly increased the bioavailability of phytic acid-bound phosphate and reduced phosphorus excretion but also diminished the antinutritional effects of phytate [5, 6]. Almost all of the phytases currently used in animal production are produced by microorganisms [7]. The ruminant gastrointestinal tract (GIT) is a rich source of phytase-producing microorganisms.

To date, many microorganisms secreting phytase have been identified in the rumen of ruminants, and some strains have been isolated and cultured in vitro for industrial production of phytase [8, 9]. Though, there are still many defects in the production and application of phytase, such as the narrow range of pH values for maximum catalytic function and poor thermal stability [10, 11]. Therefore, a large number of scientific experiments are still needed to screen more efficient and widely applicable phytase-producing microbial strains from the animal GIT. To narrow down the screening range, the most important step is to determine which microorganisms in which gastrointestinal regions have a significant correlation with dietary phosphorus digestibility. However, due to the large number of gastrointestinal microbes, traditional isolation and culture technology has not been able to complete this work. Therefore, in this study, we hypothesized that the digestibility of feed phosphorus in ruminants was influenced by the structure and composition of their gastrointestinal microbiota, and the 16S rRNA high-throughput sequencing technology was used to compare the bacterial diversity in the GIT of goats with different true digestibility of phosphorus (TDP). The results will promote a better understanding of the relationship between gastrointestinal microorganisms and host phosphorus digestibility, and lay the foundation for screening to identify high-efficiency phytase-producing strains.

## Materials and methods

The research procedure used in current study was approved by Animals policy and welfare committee of Agricultural research organization of Sichuan province China and in agreement with rules of the Animal Care and Ethical Committee of the Sichuan Agricultural University (SAU).

### Animals and sample collection

This research was conducted at the Institute of Animal Nutrition, (SAU). Twenty-eight ten months old female healthy Nubian black goats with an average initial body weight of 24.25 ± 2.47 kg were selected as research animals from the locality (Farm Panzhihua He Xie). Goats were kept in individual pens with free access to water. Total mixed ration (TMR) was provided to each goat at a restricted level of 1.1 kg DM/d (4% body weight, DM) and as shown in (Table 1) fed two times at 8:00 am and 5:00 pm. The true digestibility of dietary phosphorus was determined by a two-stage metabolic experiment according to the difference level technique (DLT) described by Ammerman [12].

The total experiment lasted for 40 days and was divided into two metabolic experiment stage (I and II). Each stage contained 20 days, which comprised of 14-day adaption duration

**Table 1. The composition and nutritional ingredients of the experimental diet (dry matter basis).**

| Ingredients | Content (%) | |
|---|---|---|
| | **Stage** | |
| | **I** | **II** |
| Alfalfa meal | 20.00 | 20.00 |
| Leymus chinensis | 35. 00 | 35. 00 |
| Corn | 38.47 | 39.15 |
| Soybean meal | 4.50 | 4.50 |
| Premix[1] | 0.45 | 0.45 |
| NaCl | 0.45 | 0.45 |
| Baking soda | 0.45 | 0.45 |
| $CaCO_3$ | 0.08 | - |
| $CaHPO_4$ | 0.60 | - |
| Total | 100.00 | 100.00 |
| Nutrition levels[2] | | |
| Metabolic energy (ME) (MJ/kg) | 9.33 | 9.46 |
| Crude protein (CP) | 9.71 | 9.79 |
| Acid detergent fiber (ADF) | 24.07 | 24.07 |
| Neutral detergent fiber (NDF) | 36.11 | 36.20 |
| Calcium (Ca) | 0.52 | 0.35 |
| Phosphorus (P) | 0.33 | 0.22 |
| Ca/P | 1.58 | 1.59 |

Premix provides the following per kg of the diet: Fe: (as ferrous sulfate) 30 mg, Cu: (as copper sulfate) 10 mg, Zn: (as zinc sulfate) 50 mg, Mn: (as manganese sulfate) 60 mg, Vitamin A: 2 937 IU, Vitamin D: 343 IU, Vitamin E: 30 IU. 2. ME is a calculated value, and the others are measured values.

and 6-day metabolism experiment duration. Except for adding $CaCO_3$ and $CaHPO_4$ (inorganic phosphorus) in the first metabolic experiment to meet the requirement of using DLT, the dietary composition and the ratio of calcium to phosphorus of the two stages were basically the same. During the metabolism trial period all the urine and feces were collected. For nitrogen fixation, the 10% of each urine and feces samples were collected each day and then mixed with 10ml of hydrochloride (10%, v/v). For nutritional composition measurement, daily feed intake and residual were recorded during the 6-day metabolism trial period and TDP was also measured on same duration.

The next day after the end of the metabolism experiment, 10 ml of blood was collected from the jugular vein of each goat using a disposable vacuum blood collection tube before feeding in the morning. After anticoagulation, the plasma was collected and stored at -20˚C until use. Then the goats were slaughtered at slaughter house in Teaching and Research Base, Animal Nutrition Institute, Sichuan Agricultural University. All the goats were slaughter according to the American Veterinary Medical Association (AVMA) guidelines for the humane slaughter of animals (The animals were stunned by captive bolt and the exsanguination from the jugular vein was carried out). The GIT organs (rumen, abomasum, jejunum, cecum and colon) were opened and the digesta was mixed and collected into tubes: rumen (8 tubes of 50 mL), abomasums (4 tubes of 15 mL), jejunum (4 tubes of 15 mL), cecum (4 tubes of 15 mL) and colon (4 tubes of 15 mL). Ruminal sample pH was measured (PHS-100 portable acidity meter, Tianqi Mdt InfoTech, Ltd., ShangHai, China) [13]; Rumen samples were filtered through four layers of cheesecloth into 3 tubes. Subsequently, this rumen fluid (approximately 20 mL) was mixed with 5mL solution of 25% metaphosphoric acid and preserved at -20˚C for

further calculation of the volatile fatty acid concentration (VFA; acetate, propionate, and butyrate). Samples from the GIT segments were stored at -80°C for further analysis.

## Sample analysis and grouping

The concentration of serum phosphorus and calcium and the activity of serum alkaline phosphatase were determined by an automatic biochemical analyzer (Automatic Analyzer 3100, Hitachi, China). The rumen liquid containing meta-phosphoric acid solution was centrifuged for 10 minutes at 12,000 × g at 4°C, the supernatant was harvested, and gas chromatography (GC-2014FRGA1, Shimadzu, and Tokyo, Japan) was used to measure the concentration of volatile fatty acids (acetate, propionate and butyrate) [14]. All feed and feces samples were dried in a forced-air oven for 48 h at 65°C, and then ground to pass a 40-mesh sieve. After that, samples were analyzed for dry matter (DM) (105°C oven, to constant weight). For determination of crude protein (CP) and ether extract (EE) the kjeldahl and Soxhlet extraction methods were used, respectively. The muffle furnace (24 h at 550°C) was used to measure the organic matter (OM) and crude ash. Neutral detergent fiber (NDF) and acid detergent fiber (ADF) were measured according to the procedures of AOAC (method NO. 973.18 C, 1990) [15]. The phosphorus content in feed and feces was determined by ammonium molybdate colorimetry [16]. The calculation of TDP presented as under:

$$TDP\ (\%) = (I_{II} - F_{II} + E)/\ I_{II} * 100$$

Where $I_{II}$ and $F_{II}$ are the phosphorus intake, and fecal phosphorus of the second metabolic experiment, respectively. E is the amount of endogenous phosphorus and was calculated as follows:

$$E = [F_{II} * I_I - (F_{I(tp)} - F_{I(ip)}) * I_{II}]/(I_I - I_{II})$$

Where $I_I$ and $I_{II}$ are the organic phosphorus intake of the first and the second metabolic experiment, respectively. $F_{I(tp)}$ and $F_{I(ip)}$ are the total phosphorus and inorganic phosphorus in the feces of the first metabolic experiment, respectively. $F_{I(ip)}$ was calculated as follows:

$$F_{I(ip)} = I_{I(ip)} * (1 - TDP_{I(ip)})$$

where $I_{I(ip)}$ and $TDP_{I(ip)}$ are the inorganic phosphorus intake and the ture digestibility of inorganic phosphorus of the first 6-day metabolic experiment period, respectively, and $TDP_{I(ip)}$ was calculated according to the principle of DLT [17] as follows:

$$TDP_{I(ip)}(\%) = (D_I - D_E)/D_I * 100$$

Where $D_I$ and $D_E$ are the difference of total phosphorus intake and fecal total phosphorus between the first and the second metabolic experiment, respectively.

TDP of all goats were calculated with mean and Standard deviation (SD). Animals were grouped into high digestibility of phosphorus (HP, TDP > mean +0.5*SD) phenotype and low digestibility of phosphorus (LP, TDP < mean—0.5*SD) phenotype, referring to the earlier defined procedure [18, 19].

## DNA extraction

Prior to DNA extraction, the content of rumen was transferred and sterile phosphate buffered saline (pH 7) was used to rinse the liquid through 4 layers of cheesecloth and collected into a sterile tube (EP). The resulting liquid was immediately centrifuged at 10,000 x g and the supernatant was gently removed. The TIANamp Bacteria DNA Kit (TIANGEN, Peking, China) was

used for DNA extraction from samples of ruminal and homogenized digesta from other organs following the manufacturer's instructions using the previously described procedure [20]. Agarose electrophoresis and NanoDrop 8000 spectrophotometer (Thermo Fisher Scientific, Brisbane, Australia) were used to measure DNA quality of the samples.

The high quality DNA was amplified using primer set 515F/860R (forward primer 515F by a sequence of 5′-GTGCCAGCMGCCGCGGTAA-3′ and primer 806R reverse by a sequence of 5′-GGACTACVSGGGTATCTAAT-3′) [21] that targets the V4 hypervariable area of the 16S rRNA bacterial gene, with a unique 5-8-base error-correcting barcode for every sample that allowed sample multiplexing in sequencing added in primer 515F end. The amplification was carried out (T100$^{TM}$ thermacycler, Bio-Rad, Hercules, CA, USA) with a ramp rate of 2.5˚C/s between all steps. The cycling parameters were: initial denaturation at 94˚C for 5 min, followed by 30 cycles (98˚C for 10 s, 50˚C for 30 s and 72˚C for 30 s), and a last extension for 5 min at 72˚C. The reaction mixture total volume was 50 μL with 200 nM each primer, dNTP mixture 5 μL of 2.5 mmol/L, Taq buffer 5 μL of 10×Ex (20 mmol/L Mg 2$^+$;TaKaRa Inc., Dalian, China), template DNA 0.35 μg, MgCl$_2$ 2mM, Taq DNA polymerase 4 units (Takara Inc., Dalian, China), and milli-Q water approximately 37 μL. The amplicons were purified with a PCR Clean-Up system (Promega, Madison, USA) by a purification kit (QIAGEN, Australia) and were quantified with a QuantiFluor™-ST fluorometer (Promega, China). Lastly, samples were sequenced on the MiSeq Illumina Sequencing Platform (Novogene Technology Co., Ltd, Beijing, China) by the method defined by Caporaso et al. [22].

## Sequencing data processing

For analyze the reads, QIIME pipeline software (version 1.8.0) was used [46]. Sequences of low-quality for example sequences containing undefined nucleotides, 3 continuous nucleotides through Q values lower than 20 and unmatched barcode sequences were detached. Usearch V7.0 founded on the Uchime algorithm implemented in QIIME [23, 24] was used to remove the chimeric sequences. Sequencing noise was additionally decreased by a preclustering procedure [25]. The Uclust [26] technique was formerly used to cluster to gained clean and high quality sequences into operational taxonomic units (OTUs) for eventual taxonomy assignment based on 97% sequence similarity (http://www.mothur.org/wiki/Greengenes-formatted_databases, gg_otu_13_8). The most abundant sequence was nominated as the representative for each OTU and was allocated to taxonomic analysis with RDP Classifier [27].

The followed chimeric OTUs were removed from the analysis against the sequence from the SILVA database [28] (http://www.mothur.org/wiki/Silva-reference-files). Alpha diversity indices (Chao 1, Shannon, Simpson and PD whole tree) and beta diversity were calculated at a sequence depth of 63,845sequences from the sample with the lowest valid sequences. Principal coordinate analysis (PCoA) was used to visualize beta diversity, as per measured with an unweighted UniFrac distance matrix [29]. According to the results of species classification, OriginPro software (version 9.0) was used to draw a relative abundance histogram of the dominant bacterial phyla. In addition, genera shared through entire samples were designated to create a heatmap with R software program version 3.4.2. All the data of sequence in current research were placed in NCBI's sequence read archive (SRA) database with the accession No. SRP185613.

## Statistical analysis

All of the Statistical analysis of data was conducted using SPSS Statistics software v. 19.0 (IBM, Armonk, NY, USA). The normality of data was investigated with a Shapiro–Wilk test prior to all statistical analyses. According to the results of normality investigation, the difference of the

relative bacterial abundance between the HP and LP groups was analyzed by a nonparametric test of two samples, and other parameters were analyzed by unpaired t-test. Spearman rank correlation analysis was performed to assess the correlation between TDP and the relative abundance of bacteria. The results were presented as the means ± SD, and the significant and extremely significant levels were set at P < 0.05 and P < 0.01, respectively.

## Results

### Comparison TDP between group HP and LP

The TDP of the goats varied from 68.4% to 90.3% with an average of 83.3% (± 6.8%). In present research, 8 individuals were collected in each group of LP and HP phenotype respectively. The TDP of the 8 animals in the HP group (89.5% ± 2.9%) was significantly higher than the LP group (78.4% ± 1.21%).

### Comparison of blood parameters, ruminal fermentation parameters and apparent digestibility of nutrients between group HP and LP

The serum phosphorus content of the HP group was significantly higher (P<0.05) than the LP group, was significant greater compare to LP group (P < 0.05), while the serum calcium content, rumen fermentation parameters, rumen pH and volatile fatty acids (acetate, propionate, butyrate and acetic/propionic acid), and the apparent digestibility of nutrients (DM, EE, CP, ADF and NDF) were not significantly different (P > 0.05) between the groups and presented in (Tables 2–4).

### Data acquired from sequencing

16S rRNA sequencing analysis, a high quality sequences, total of 5,711,204 were produced, in rumen 915,084, in abomasum 931,464, in jejunum 900,688, in colon 909,122, and in cecum 813,496. At 97% nucleotide sequence identity amid reads, 5,777 OTUs were recognized in rumen, in abomasum 5,810, in jejunum 5,611, in colon 8,886 and in cecum 6,883, with an average of 2052±74 and 2246±81 per sample in the rumen of the HP and LP groups, 2086±186 and 2178±105 in the abomasum, 2313±179 and 2085±271 in the jejunum, 3052±374 and 2848 ±255 in the colon, and 2609±358 and 2948±163 in the cecum, respectively (S1 Table). The number of shared OTUs between groups was, in rumen 1,868, in abomasum 1,760, in jejunum 2,196, in colon 2,067 and in cecum 2,148 and presented in (Fig 1A). A total number of 140,403 OTUs were identified in the present work. A rarefaction curve analysis for the OTUs was performed and the results were presented in Fig 1B. All of the rarefaction curves reached the saturation level, which indicated that the sequencing depth of the present study captured the majority of bacteria across the GIT.

### Alpha diversity and beta diversity analysis

The fruitfulness and consistency of the GIT microbiota across gut as specified by the alpha diversity indices were evaluated. As shown in Fig 2A, the Shannon and Simpson indices in the

**Table 2. Comparison of blood parameters between the HP and LP groups.**

| Item | HP (n = 8) | LP (n = 8) | P value |
| --- | --- | --- | --- |
| Ca (mmol/L) | 2.17±0.22 | 2.13±0.12 | 0.647 |
| P (mmol/L) | 2.27±0.23 | 2.01±0.22 | 0.034 |
| Alkaline phosphatase of blood (U/L) | 160.75±26.85 | 162.12±27.19 | 0.920 |

**Table 3. Comparison of rumen fermentation parameters between the HP and LP groups.**

| Item | HP (n = 8) | LP (n = 8) | P value |
|------|------------|------------|---------|
| Rumen pH | 6.24±0.24 | 6.31±0.21 | 0.582 |
| Acetate (mmol/L) | 32.24±7.34 | 30.24±3.81 | 0.505 |
| Propionate (mmol/L) | 10.32±1.73 | 10.58±0.98 | 0.720 |
| Butyrate (mmol/L) | 7.95±1.83 | 8.24±1.59 | 0.738 |
| Acetate/Propionate | 3.17±0.78 | 2.86±0.38 | 0.338 |

rumen of LP were significant (p < 0.05) greater compare to those of HP, and the Simpson index in the abomasum of LP was significant (p < 0.05) greater compare to HP. In addition, the Chao 1 index in the jejunum of HP was significant (p < 0.05) greater compare to LP. The differences in other alpha diversity indices amid the groups across the GIT were not significant (p > 0.05) (Fig 2A).

Beta diversity measurements of the GIT microbiome across gut were displayed with a plot (PCoA) through un-weighted UniFrac metrics distance (Fig 2B). A nearer distance between two points designated samples higher similarity, and the percentage of variation was explained by PC1 and PC2, as directed by the axis. In the present research, PC1 and PC2 described variation of 52.63%, and 15.547% respectively. The PCoA plot displayed no noticeable borderline of microbiota between HP and LP across the GIT. The samples were clustered according to the region of the GIT they were from rather than the group they were in. On the PCoA plot, the samples of the rumen and abomasum gathered together and separated from the large and small intestinal samples along the X axis, and the samples of the large intestine (colon and cecum) gathered together and separated from the small intestinal samples along the Y axis. A plot of analysis of similarities (ANOSIM) (Fig 2C) also displayed non-significant difference in microbial communities between the groups. Generally, the similarities of microbial communities originating from the same/adjacent GIT segments were higher than those from other regions.

## Bacterial composition of the gastrointestinal tract

In phylum level, 23,30,36,23 and 25 taxa were recognized in rumen, abomasum, jejunum, cecum and colon respectively. (Fig 3A) displays the compositions of the top 10 plentiful phyla in each group. The dominant bacteria in rumen and abomasum were similar, with the most abundant phylum being Bacteroidetes (HP 62.84% and LP 58.43% in the rumen; HP 45.10% and LP 44.28% in the abomasum) and the second most abundant being Firmicutes (HP 29.44% and LP 33.20% in the rumen; HP 29.44% and LP 33.20% in the abomasum). In colon, cecum and jejunum the leading bacteria were similar, with the utmost plentiful phylum being Firmicutes (HP 66.69% and LP 62.22% in the jejunum; HP 66.22% and LP 59.35% in the colon; HP 64.92% and LP 54.05% in the cecum), and the second most abundant bacteria were Bacteroidetes (HP 6.63% and LP 4.44% in the jejunum; HP 19.10% and LP 21.07% in the colon; HP 20.79% and LP 18.07% in the cecum).

**Table 4. Comparison of apparent digestibility of nutrients between the HP and LP groups (%).**

| Item | HP (n = 8) | LP (n = 8) | P value |
|------|------------|------------|---------|
| Dry matter (DM) | 48.94±4.81 | 48.92±3.33 | 0.992 |
| Ether extract (EE) | 39.56±6.56 | 45.21±6.17 | 0.098 |
| Crude protein (CP) | 48.33±4.62 | 49.47±7.91 | 0.730 |
| Acid detergent fiber (ADF) | 48.77±5.36 | 44.91±4.75 | 0.479 |
| Neutral detergent fiber (NDF) | 50.04±2.37 | 48.21±4.53 | 0.350 |

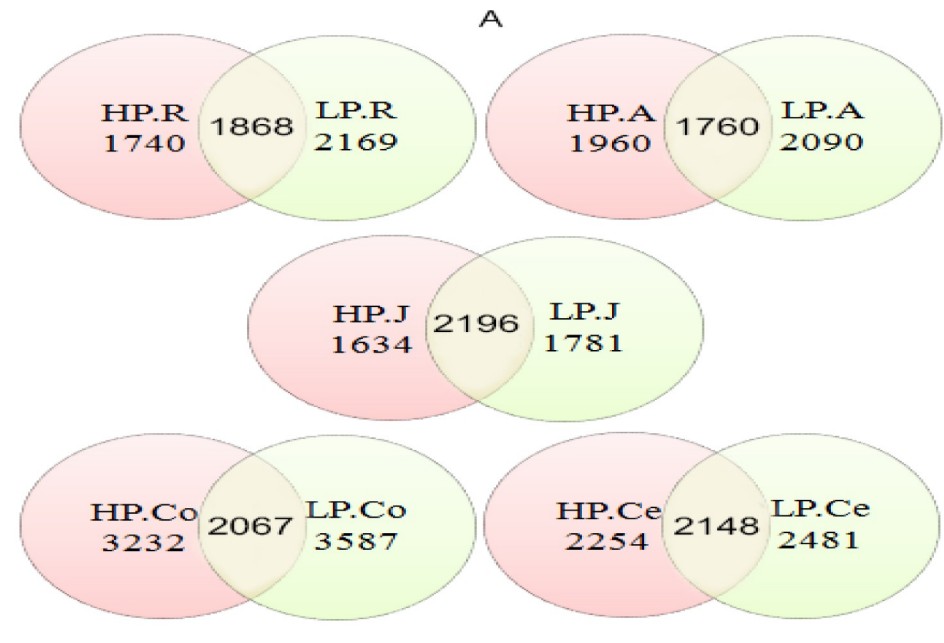

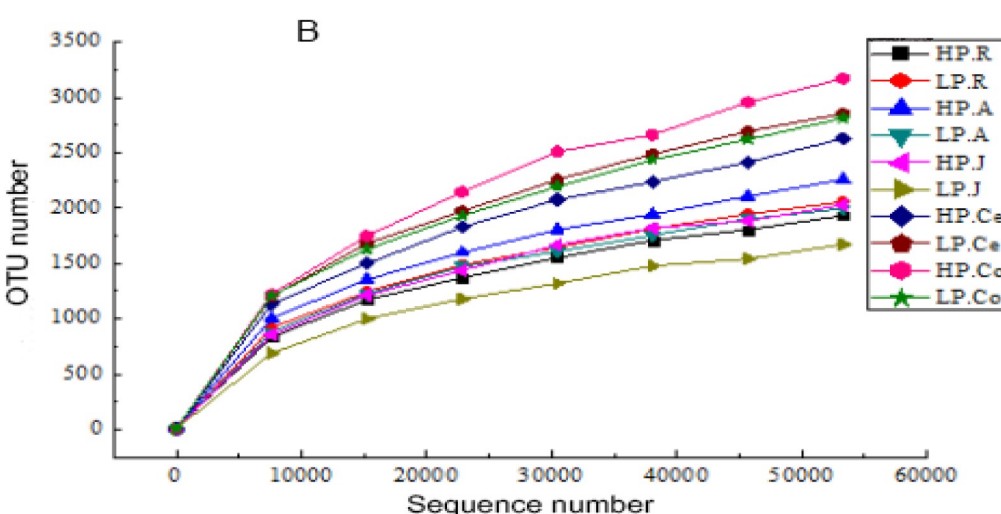

**Fig 1.** A: Venn diagram representation of the shared and exclusive OTUs. B: Rarefaction curves of each group. The rarefaction analysis was conducted at 97% sequence identity, and each curve represented one individual gastrointestinal sample of goats (n = 8). HP.R: rumen of HP; LP.R: rumen of LP; HP.A: abomasum of HP; LP.A: abomasum of LP; HP.J: jejunum of HP; LP.J: jejunum of LP; HP.Co: colon of HP; LP.Co: colon of LP; HP.Ce: cecum of HP; LP.Ce: cecum of LP.

In rumen, abomasum, jejunum, colon and cecum a total of 239,319,248,248 and 246 genera were detected respectively. Amid the acquired sequences, while an average 38.19%, 29.93%, 23.45%, 27.25% and 26.53% in rumen, abomasum, jejunum, colon and cecum were not recognized at the genus level. The average relative abundances of the top 20 shared genera are displayed in a heatmap (Fig 3B). In the rumen the dominant genera were *Prevotella_1* (HP 15.04%, LP 17.19%) and *Rikenellaceae_RC9_gut_group* (HP 10.97%, LP 11.37%). In the abomasum, *Prevotella_1* (HP 17.31%, LP 13.49%) and *Rikenellaceae_RC9_gut_group* (HP

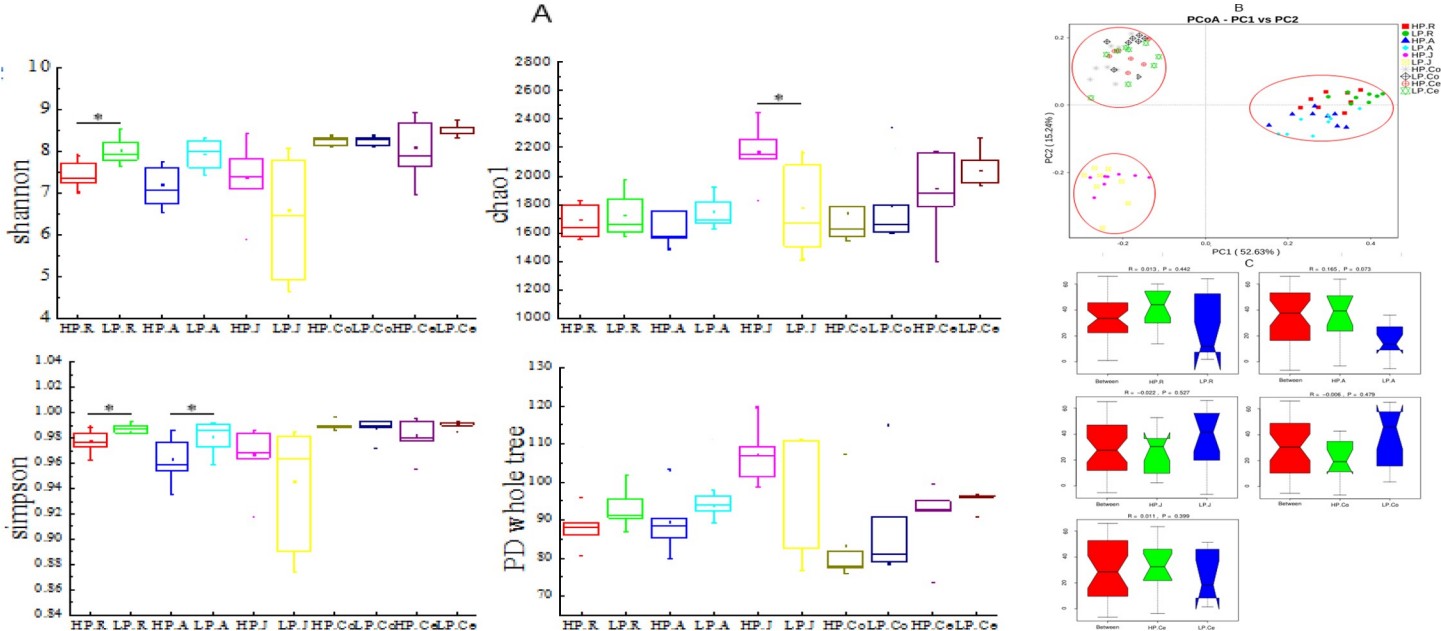

**Fig 2.** A: Comparison of alpha diversity indices between the HP and LP groups. Error bars represent the SD of two samples. Asterisks show significant differences between groups (*P<0.05), the same as below. B: Principal coordinate analysis of bacterial samples from the gastrointestinal tract. A greater distance between two samples indicated a lower similarity. The percentage of variation explained by PC1 and PC2 are indicated on the axis. C: A microbial analysis of similarities (ANOSIM). R was the mean rank of between-group dissimilarities, ranging from -1 to 1. When the R value is greater than 0, the difference between groups is greater than that within groups, indicating that there are differences between groups; when the P value is less than 0.05, it indicates significant differences between groups. HP.R: rumen of HP; LP.R: rumen of LP; HP.A: abomasum of HP; LP.A: abomasum of LP; HP.J: jejunum of HP; LP.J: jejunum of LP; HP.Co: colon of HP; LP.Co: colon of LP; HP.Ce: cecum of HP; LP.Ce: cecum of LP.

4.51%, LP 6.54%) were the two utmost plentiful genera, alike to the rumen. In jejunum, colon and cecum the dominant genera were very dissimilar since *Christensenellaceae_R-7_group* (HP 22.81%, LP 9.03%) and *Ruminococcaceae_NK4A214_group* (HP 3.21%, LP 2.77%) were the utmost plentiful genera in the jejunum, while *Rikenellaceae_RC9_gut_group* (HP 7.93% and LP 6.21% in the colon; HP 8.15% and LP 5.54% in the cecum) was the primary genus both in the colon and cecum.

## Compression of bacterial composition across the gastrointestinal tract between the HP and LP

A significant difference between the HP and LP groups were observed, when the relative abundance in phylum to genus level were compared (Table 5). At the phylum level, in the abomasum, the relative abundance of Firmicutes was significant (p < 0.05) higher, while Lentisphaerae was significantl (p<0.05) lower in the LP compare to HP group. In the colon, the relative abundances of Verrucomicrobia, Lentisphaerae and Firmicutes were significant (p < 0.05) higher in the LP compare to HP group. At the genus level, in the rumen, the relative abundances of *Ruminococcaceae_UCG-010*, *Ruminococcus_2*, *Ruminococcaceae_NK4A214_group*, *Saccharofermentans*, *Ruminococcaceae_UCG-014*, *Mogibacterium; Eubacterium_nodatum_group* and *Family_XIII_UCG-002* were significant lower (*P* < 0.05) in the HP compare to LP group, although *Selenomonas_1*, *Prevotella* and *Treponema_2* were significant greater (*P* < 0.05) in the HP compare to LP group. In the abomasum, *Rikenellaceae_RC9_-gut_group*, *Lachnospiraceae_ND3007_group*, *Saccharofermentans*, *Ruminococcus_1*, *Eubacterium_coprostanoligenes_group*, *Ruminococcaceae_UCG-014*, *Lachnospiraceae_XPB1014_group*,

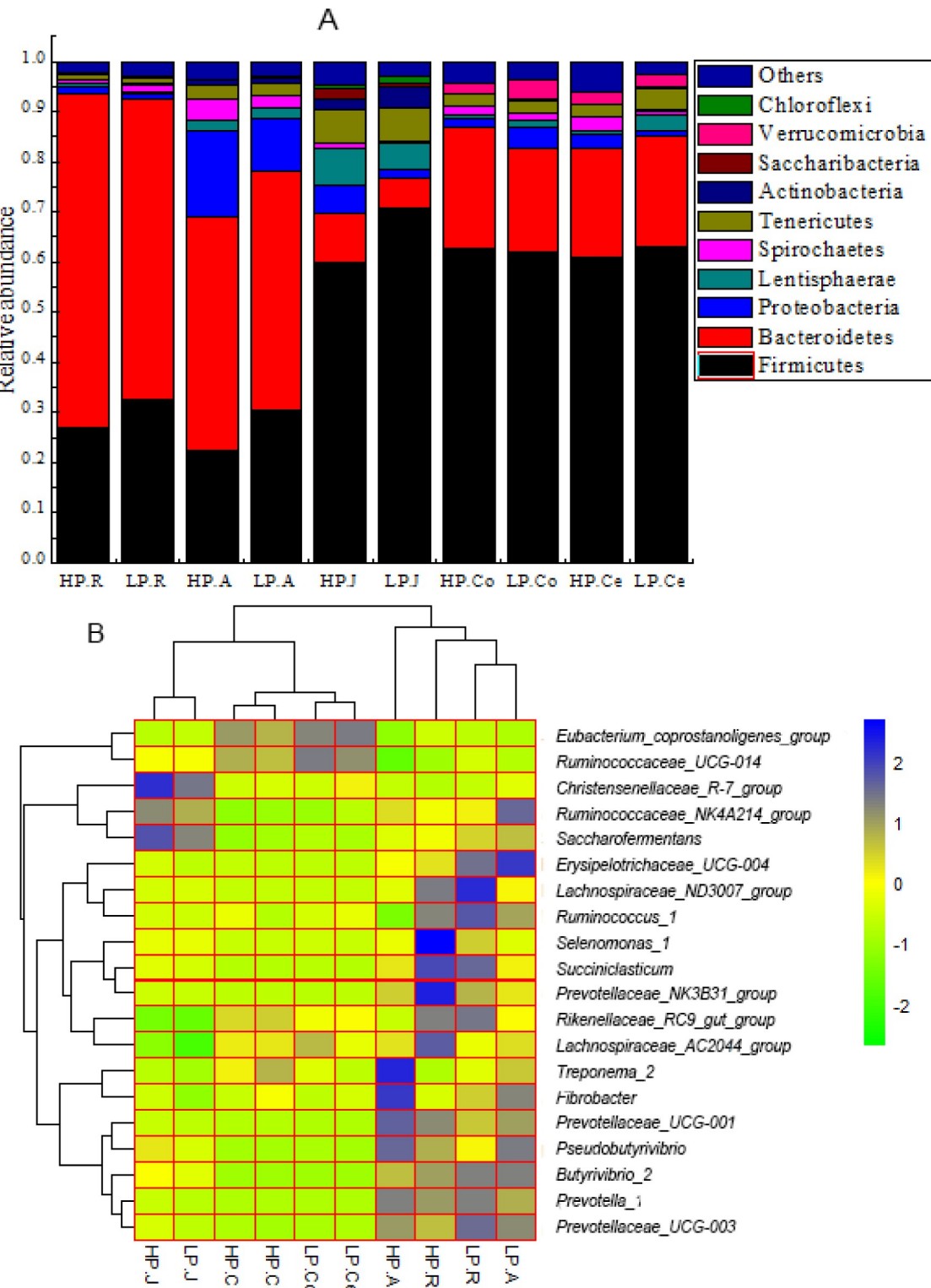

**Fig 3.** A: Bacterial compositions across the gastrointestinal tract at the phylum level (only the top 10 abundant phyla are presented). HP.R: rumen of HP; LP.R: rumen of LP; HP.A: abomasum of HP; LP.A: abomasum of LP; HP.J: jejunum of HP; LP.J: jejunum of LP; HP.Co: colon of HP; LP.Co: colon of LP; HP.Ce: cecum of HP; LP.Ce: cecum of LP. B: Heatmap of the core genera across the gastrointestinal tract (the relative abundance of microbes was log-transformed, and only the top 20 abundant genera are presented). The closer to the color blue, the higher the relative abundance, while the closer to the color green, the lower the relative abundance.

**Table 5. The bacterial phyla and genera whose relative abundances were significantly different between HP and LP across the GIT.**

| Taxa | | | Relative abundance (>0.1%) | | |
|---|---|---|---|---|---|
| | | | LP | HP | P value |
| Rumen | | | | | |
| Genus | | Ruminococcaceae_UCG-010 | 0.702±00.186 | 0.459±0.140 | 0.029 |
| | | Selenomonas_1 | 0.225±0.105 | 0.398±0.106 | 0.018 |
| | | Treponema_2 | 0.481±0.103 | 0.713±0.084 | 0.002 |
| | | Prevotella | 17.0±0.571 | 18.9±0.753 | 0.001 |
| | | Ruminococcus_2 | 0.609±0.226 | 0.245±0.141 | 0.008 |
| | | Ruminococcaceae_NK4A214_group | 2.07±0.555 | 1.39±0.330 | 0.028 |
| | | Saccharofermentans | 1.06±0.204 | 0.652±0.296 | 0.019 |
| | | Ruminococcaceae_UCG-014 | 0.901±0.202 | 0.568±0.032 | 0.003 |
| | | Mogibacterium | 0.131±0.066 | 0.064±0.021 | 0.039 |
| | | Eubacterium_nodatum_group | 0.106±0.045 | 0.062±0.021 | 0.036 |
| | | Family_XIII_UCG-002 | 0.108±0.045 | 0.055±0.031 | 0.042 |
| Abomasum | | | | | |
| Phylum | | Firmicutes | 30.7±6.29 | 22.6±5.78 | 0.038 |
| | | Lentisphaerae | 0.151±0.075 | 0.367±0.122 | 0.001 |
| Genus | | Rikenellaceae_RC9_gut_group | 6.57±0.959 | 4.50±1.965 | 0.043 |
| | | Lachnospiraceae_ND3007_group | 0.662±0.219 | 0.276±0.110 | 0.003 |
| | | Saccharofermentans | 1.17±0.347 | 0.540±0.222 | 0.004 |
| | | Ruminococcus_1 | 1.14±0.289 | 0.569±0.101 | 0.001 |
| | | Eubacterium_coprostanoligenes_group | 0.676±0.234 | 0.332±0.147 | 0.012 |
| | | Ruminococcaceae_UCG-014 | 0.755±0.219 | 0.375±0.112 | 0.004 |
| | | Lachnospiraceae_XPB1014_group | 0.628±0.220 | 0.384±0.122 | 0.039 |
| | | Desulfovibrio | 0.216±0.064 | 0.128±0.043 | 0.020 |
| | | Lachnoclostridium_10 | 0.230±0.181 | 0.045±0.016 | 0.033 |
| | | Anaerovorax | 0.261±0.049 | 0.150±0.049 | 0.003 |
| | | Papillibacter | 0.243±0.078 | 0.123±0.034 | 0.006 |
| | | Ruminococcaceae_UCG-004 | 0.240±0.083 | 0.111±0.053 | 0.010 |
| | | Oscillospira | 0.322±0.167 | 0.151±0.073 | 0.044 |
| Jejunum | | | | | |
| Genus | | Prevotella | 1.92±0.911 | 5.34±2.66 | 0.014 |
| | | Rikenellaceae_RC9_gut_group | 1.58±0.25 | 2.640±0.296 | <0.001 |
| | | Fibrobacter | 0.376±0.173 | 0.825±0.260 | 0.006 |
| | | Desulfovibrio | 0.058±0.032 | 0.134±0.019 | 0.001 |
| | | Victivallis | 3.56±2.33 | 0.724±0.199 | 0.014 |
| | | Ruminococcus_2 | 3.39±0.276 | 2.47±0.431 | 0.001 |
| | | Clostridium_sensu_stricto_1 | 0.851±0.119 | 5.08±0.559 | <0.001 |
| Colon | | | | | |
| Phylum | | Verrucomicrobia | 5.07±2.57 | 2.41±1.40 | 0.049 |
| | | Lentisphaerae | 0.451±0.092 | 0.299±0.105 | 0.024 |
| | | Firmicutes | 66.2±4.81 | 59.4±7.38 | 0.045 |
| Genus | | Victivallis | 0.461±0.096 | 0.305±0.105 | 0.024 |
| | | Ruminococcaceae_UCG-014 | 1.76±0.119 | 1.53±0.134 | 0.010 |
| Cecum | | | | | |
| Genus | | Desulfovibrio | 0.226±0.065 | 0.373±0.081 | 0.010 |

*Desulfovibrio, Lachnoclostridium_10, Anaerovorax, Papillibacter, Ruminococcaceae_UCG-004* and *Oscillospira* had significant difference ($P < 0.05$) relative abundances between the groups.

In the jejunum, the relative abundances of *Prevotella*, *Rikenellaceae_RC9_gut_group*, *Fibrobacter*, *Desulfovibrio* and *Clostridium_sensu_stricto_1* were significant greater ($P < 0.05$), while *Victivallis* and *Ruminococcus_2* were significant lower ($P < 0.05$), in the HP group compare to the LP group. In the colon, the relative abundances of *Victivallis* and *Ruminococcaceae_UCG-014* were significant higher (p < 0.05) in the LP group compare to the HP group. In the cecum, *Desulfovibrio* was significant (p<0.05) greater in the HP compare to the LP animals.

## Correlation between microorganisms and TDP

To observe the association between the bacterial relative abundance and TDP in every section of the GIT, analysis of correlation was done. The results (Fig 4) displayed that in the rumen, TDP was negatively involved in the relative abundance of Erysipelotrichaceae (family), *Ruminococcaceae_UCG-010*, *Ruminococcus_2* and *Ruminococcaceae_UCG-014* but positively involved in the relative abundance of *Selenomonas_1* and *Prevotella*. In the abomasum, two phyla, one class, two orders, six families and thirteen genera were predisposed by the variation of the TDP. In the jejunum, TDP was positively correlated with the relative abundances of six genera (*Desulfovibrio*, *Fibrobacter*, *Prevotella*, *Rikenellaceae_RC9_gut_group*, *Ruminococcus_2* and *Victivallis*) and negatively correlated with one genus (*Clostridium_sensu_stricto_1*). In the colon, TDP was negatively involved in the relative abundance of three phyla, two classes, two orders, one family and two genera. In the cecum, at the genus level, only *Desulfovibrio* was positively correlated with TDP.

## Discussion

Due to the presence of endogenous phosphorus in the saliva and digestive tract of ruminants, apparent digestibility seriously underestimated the true digestion and absorption of feed phosphorus [17, 30]. Therefore, only TDP can reflect the digestion efficiency of dietary phosphorus. The endogenous phosphorus outputs and TDP values for animals are usually detected using regression analysis technique (REG) and DLT, and the latter was adopted in this study. In current research, the phosphorus dietary levels in the first and second metabolic experiments were 2.2 and 3.3 g/kg, respectively, which were lower than the recommended value of the Chinese meat sheep and goat feeding standard (4.0–5.0 g/kg DMI, NY/T 816–2004). In addition, the raw materials of the experimental diet of the two stages were basically the same. These are the main requirements of using the DLT to determine TDP. The average TDP of this study was 83.28%, which was higher as compared to pigs [31] or chickens [32]. Compared with

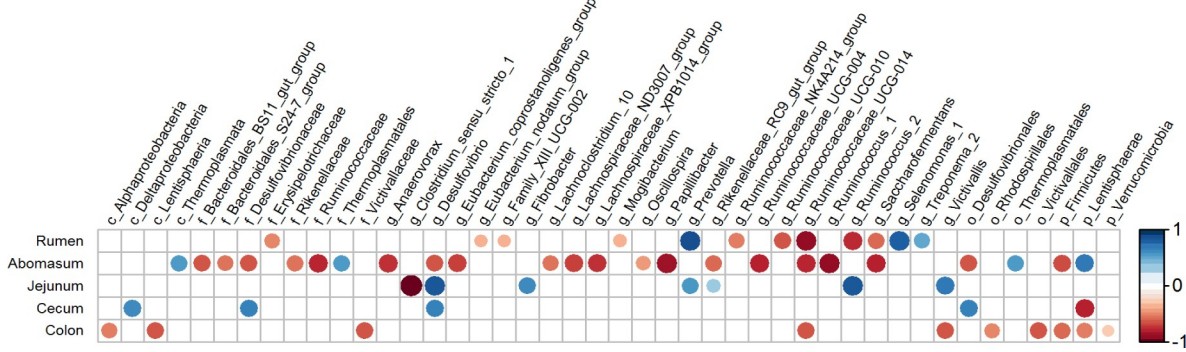

**Fig 4. Correlation analysis between the bacteria across the gastrointestinal tract and the true digestibility of phosphorus in goats.** Correlations are indicated by large circles and weaker correlations by small circles. The scale colors denote whether the correlation is positive (closer to 1, red circles) or negative (closer to −1, blue circles) between the taxa and gastrointestinal tract variables.

monogastric animals, the digestibility of feed phosphorus was higher in ruminants, mainly because ruminants have more abundant gastrointestinal microorganisms that can secrete phytase to hydrolyze phytic acid to release phosphorus, thereby promoting the digestion of feed phosphorus [33].

Earlier research showed that the digestion efficiency of feed phosphorus was mainly affected by animal and dietary factors, which included the animal species and age, the source of feed phosphorus, the phosphorus level and the ratio of calcium to phosphorus [34, 35]. The current research displayed that the TDP of twenty-eight Nubian goats varied greatly from 68.38% to 90.26%, even though the feeding management, genetic background and age of goats were similar. This phenomenon may have been observed by researchers in previous research activities, but prior to our research, there was no published report on the variation of TDP of livestock. Nevertheless, earlier research showed that individual variations in feed efficiency of ruminants are common [36–38]. TDP is one of the vital components of feed efficiency; therefore, the individual variations of TDP should also be reasonable.

According to our knowledge, the present study was the first experiment evaluating the influence of the composition and structure of gastrointestinal bacteria on the host TDP. The results showed that the number of bacterial genera significantly associated with host TDP was the highest in the abomasum (thirteen), followed by the rumen (eleven), the jejunum (seven), the colon (two) and the cecum (one). This result indicated that in stomach and small intestine the bacteria, rather than in the large intestine, played a vital role in influencing phosphorus digestibility. It is well known that the rumen and jejunum are important locations for ruminants to digest nutrient, and microbial digestion is an important component of digestion in the ruminant digestive tract. Therefore, the significant correlation between microbes in the rumen and jejunum and feed phosphorus digestibility is easy to understand. The role of the abomasum in feed digestion has always been weak. It was unexpected that a large number of bacteria in the abomasum also were found to be significantly associated with host TDP in the present study. The positive correlation between gastrointestinal bacteria and host TDP should be attributed to the presence of phytase production bacteria, because earlier studies found that gastrointestinal bacteria of ruminants, such as Selenomonas and Prevotella [39], Bifidobacterium [40], and Shewanella [41], were able to secrete phytase to hydrolyze phytic acid and improve the digestibility of feed phosphorus [42, 43]. The negative correlation indicated that some gastrointestinal microbes may inhibit the digestion of feed phosphorus, and the mechanism has not yet been reported. It was speculated that the secretion of those bacteria can catalyze the binding of free phosphorus with some substance in feed such as protein and fat, which were not easily digested and thus inhibited the digestion and absorption of phosphorus [44]. It might also be that there was a competitive or antagonistic relationship between those bacteria and phytase-producing species, and their large-scale reproduction can inhibit the growth of phytase-producing microorganisms, thereby reducing the production of phytase in the GIT.

This study found that some bacterial genera, even those located in different segments of the GIT, had the same correlation with TDP. For example, *Prevotella* was significantly positively correlated with TDP in both the rumen and the jejunum, and *Ruminococcaceae_UCG-014* in the rumen, abomasum and colon was negatively correlated with TDP. This result indicated that the characteristics related to phosphorus digestion of these bacteria were relatively stable. However, some bacteria exhibited opposite correlations with TDP when they were located in different parts of the GIT. *Ruminococcus_2*, for example, was negatively correlated with host TDP in the rumen but positively correlated with TDP in the jejunum; *Desulfovibrio* was negatively correlated with TDP in the rumen but positively correlated with TDP in the jejunum and cecum. We speculated that there may be two reasons for this phenomenon. First, the biological characteristics of these bacteria might be susceptible to environmental impact. After all,

the internal environment of different parts of the GIT, such as pH value, osmotic pressure and nutrient availability is different [45, 46]. The second reason might be related to the limitations of high-throughput sequencing technology, which can only annotate OTUs accurately to the microbial genus level, not species. Each bacterial genus consists of many species, and different species have different biological functions. For example, in the genus *Desulfovibrio*, there are at least 62 species and eight subspecies [47], which are usually present in the animal GIT, microbial mats, marshes, soda lakes, anoxic mud and marine sediments, and they are highly divergent on the basis of their phylogenetic and phenotypic characteristics [48].

Besides being relate to the ability of rumen microorganisms to produce phytase, the correlation between gastrointestinal bacteria and TDP in goats found in this study may also be related to the difference in rumen microbial protein synthesis between groups. Previous research found that the ratio of nitrogen to phosphorus in ruminal microbial cell mass is rather constant [49], differences in microbial protein synthesis would result in changes of true phosphorus digestibility. Unfortunately, this study did not measure the activity of microbial phytase in the GIT and the amount of microbial protein synthesis in the rumen. Therefore, it was uncertain that the improvement of feed phosphorus digestibility was achieved through the increases of microbial phytase secretion or microbial protein synthesis. It was noteworthy that the attention of this study was focused on the bacteria, while the relationship between other microorganisms, such as fungi, and host TDP was ignored. Previous studies have shown that phytase-secreting fungi exist in both the rumen [39] and soil [36]. The correlation between rumen fungi and host TDP remains unclear and is worthy of further study.

## Conclusions

In summary, there were obvious individual variations in the TDP of goats. Furthermore, the structure of goat's gastrointestinal microbiota differing in TDP was distinctly dissimilar. Microbes in rumen and abomasum had the utmost effect on host TDP then those of other segments of gastrointestinal tract. Some gastrointestinal bacteria, such as ruminal *Prevotella*, were beneficial for host true digestibility of feed phosphorus.

## Supporting information

**S1 Table. The numbers of OTUs and reads.**
(PDF)

## Acknowledgments

We thank Kaizhen Liu for extraction and amplification of gene. We also thank Lei Lao for collection of samples and data analysis.

## Author Contributions

**Conceptualization:** Lizhi Wang, Zhisheng Wang.

**Data curation:** Lizhi Wang, Zhisheng Wang.

**Formal analysis:** Lizhi Wang, Ali Mujtaba Shah, Lei Jin, Bai Xue.

**Funding acquisition:** Lizhi Wang.

**Investigation:** Ali Mujtaba Shah, Bai Xue.

**Methodology:** Ali Mujtaba Shah, Yuehui Liu, Lei Jin, Quanhui Peng.

**Project administration:** Lizhi Wang.

**Software:** Ali Mujtaba Shah, Yuehui Liu, Lei Jin, Quanhui Peng.

**Supervision:** Zhisheng Wang.

**Validation:** Quanhui Peng.

**Writing – original draft:** Lizhi Wang, Ali Mujtaba Shah.

**Writing – review & editing:** Ali Mujtaba Shah.

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
