## [Decision Letter · Decision Letter 0]

20 Jan 2020

PONE-D-19-29759

Relationship between True Digestibility of dietary Phosphorus and Gastrointestinal Bacteria of Goats

PLOS ONE

Dear Dr. Wang,

Thank you for submitting your manuscript to PLOS ONE. After careful consideration, we feel that it has merit but does not fully meet PLOS ONE’s publication criteria as it currently stands. Therefore, we invite you to submit a revised version of the manuscript that addresses the points raised during the review process.

The reviewers were split on their opinion of the manuscript.  In addition to a number of minor edits, there is a recommendation to strengthen and clarify the hypothesis and goals of the manuscript, all of which I agree with.  I would like to give this opportunity to the authors to comment and perhaps revise their paper.  If the authors choose to submit a revision, I may solicit the opinion of a third reviewer. 

We would appreciate receiving your revised manuscript by Mar 02 2020 11:59PM. To enhance the reproducibility of your results, we recommend that if applicable you deposit your laboratory protocols in protocols.io, where a protocol can be assigned its own identifier (DOI) such that it can be cited independently in the future. For instructions see: http://journals.plos.org/plosone/s/submission-guidelines#loc-laboratory-protocols

We look forward to receiving your revised manuscript.

Kind regards,

Suzanne L. Ishaq, PhD

Academic Editor

PLOS ONE

Journal Requirements:

2. In your Methods section, please provide additional information on the animal research and ensure you have included details on : (1) methods of blood collection, e.g. volume, location from which samples were taken, methods of anesthesia if relevant (2) methods of goat sacrifice, and (3) efforts to alleviate suffering.

Reviewers' comments:

Reviewer's Responses to Questions

**Comments to the Author**

1. Is the manuscript technically sound, and do the data support the conclusions?

Reviewer #1: No

Reviewer #2: Yes

2. Has the statistical analysis been performed appropriately and rigorously? 

Reviewer #1: Yes

Reviewer #2: Yes

3. Have the authors made all data underlying the findings in their manuscript fully available?

Reviewer #1: Yes

Reviewer #2: Yes

4. Is the manuscript presented in an intelligible fashion and written in standard English?

Reviewer #1: No

Reviewer #2: Yes

5. Review Comments to the Author

Reviewer #1: The present aimed at characterizing the interrelation between true digestibility of phosphorus and gastrointestinal bacteria in goats. Therefore, the animals were kept at two different levels of P and at the end of the metabolic trials the animals were slaughtered and samples for characterizing the microbiome along the gastrointestinal axis were taken.

Two major comments have to be guiven:

1. the differences in dietary P supply were marginal , i. e., according to plasma Ca and P concentrations the dietary treatment cannot be assigned as "LP".

The whole study is not hypothesis driven. Thus, a clear concept is lacking.

Reviewer #2: Line comment

28 phyla

49 , contributing to surface water eutrophication in locations where there is intensive monogastric livestock production [2,3]. Phytase is an enzyme that

67 technology has not been able to accomplish this work. Therefore, this study with 16S rRNA….

69 true digestibilities of phosphorus

71 screening to identify high-efficiency….

80 old healthy Nubian black goats with an average…

82 Goats were kept in individual pens with free access to water.

84 ) as shown in (Table 1) fed two times…

86 delete sentence starting with The formulas

90 Vitamin A, Vitamin D and Vitamin E is conventional nomenclature

97 collected. Was urine collected into acid? Or just mixed with acid after sampling?

97 An aliquot, representing 10% of the daily output was collected each day, mixed with 10 ml of 10% hydrochloric acid and composited.

100 6-day metabolism

103 jejunum…) were opened and digesta mixed and collected into tubes: rumen (8 tubes of 50 ml) ….

105 Ruminal sample pH was measured (PHS….

107]. Ruminal fluid was strained through four layers of cheese cloth and 20 ml were mixed with 4 mL of 25%? Meta-phosphoric acid…

110 Samples from the GIT segments were stored at -80oC for further analysis.

116 gas chromatography

119 ground to pass a 40-mesh sieve

122 measure organic matter……crude ash. Neutral detergent fiber…What about mentioning specific procedure from AOAC book?

124 without sodium sulfite and corrected for residual ash.

136 true digestibility

142 TDP of all goats were calculated with mean…

142 Animals were grouped into high digestibility of phosphorus….

147 and sterile phosphate buffered saline (pH 7) was used to rinse the liquid through 4 layers of cheesecloth and collected into a sterile tube (EP). The resulting liquid was immediately centrifuged at 10,000 x g and the supernatant gently removed.

152 used for DNA extraction from samples of ruminal and homogenized digesta from other organs following the manufacturers instructions using the procedure described in [19]

154 Agarose electrophoresis and NanoDrop

155 measure DNA quality of the samples.

184 SILVA

204 The TDP of the goats varied from 68.4% to 90.3% with an average of 83.3% (± 6.8%).

205 delete sentence.

206 The TDP of the 8 animals in the HP group 89.5% ± 2.9%) was significantly higher (than for the LP group (78.4% ± 1.21%).

209 was significantly higher (P<.05) than the LP group

212 not signficantly different (P>0.05)

230-359 While I recognize some phyla and genus, I am not familiar enough with literature to give a critical review. Yes, I have seen these in other research, but not knowledgeable enough to evaluate the research.

368 what is the feeding standard recommendation?

379 goats varied greatly.

382 but prior to our research, there was no published report on the variation of TDP…

383 earlier research showed that individual variations in feed efficiency of ruminants are common {…

393 microbial digestion is an important component of digestion in the

404 yet been reported.

405 the binding of free phosphorus with some substance in feed such as protein…

421 nutrient availability

445 CABI

451 feeding

504 excretion in

Why not in figure 1B and 3a CHANGE SYMBOLS. Use a solid square for HP rumen and an open square for LP rumen. Use a solid triangle for the abomasum HP and an open triangle for LP. Use a solid and open circle for jejunum and open and closed diamond for the colon.

6. PLOS authors have the option to publish the peer review history of their article (what does this mean?). If published, this will include your full peer review and any attached files.

Reviewer #1: No

Reviewer #2: Yes: Steve Hart

---

## [Author Response · Author response to Decision Letter 0]

6 Feb 2020

Editor: Please ensure that your manuscript meets PLOS ONE's style requirements, including those for file naming. 

Response:We download the PLOS ONE style templates from the website, and formatted our manuscript as requested.

Editor: In your Methods section, please provide additional information on the animal research and ensure you have included details on: (1) methods of blood collection, e.g. volume, location from which samples were taken, methods of anesthesia if relevant (2) methods of goat sacrifice, and (3) efforts to alleviate suffering.

Response: we have revised our manuscript and the related contents were presented in L105-112 as “The next day after the end of the metabolism experiment, 10 ml of blood was collected from the jugular vein of each goat using a disposable vacuum blood collection tube before feeding in the morning. After anticoagulation, the plasma was collected and stored at -20 °C until use. Then the goats were slaughtered. Pre-harvest handling was conducted in accordance with good animal welfare practices, and slaughtering procedures followed actual chinese law on animal production and sanitary inspection.”

Editor: In your Data Availability statement, you have not specified where the minimal data set underlying the results described in your manuscript can be found. PLOS defines a study's minimal data set as the underlying data used to reach the conclusions drawn in the manuscript and any additional data required to replicate the reported study findings in their entirety. All PLOS journals require that the minimal data set be made fully available.

Response: In the “Data Availability Statement”, we state that “All the data of sequence in current research were placed in NCBI's sequence read archive (SRA) database with the accession No. SRP185613.”

Editor: PLOS requires an ORCID iD for the corresponding author in Editorial Manager on papers submitted after December 6th, 2016. Please ensure that you have an ORCID iD and that it is validated in Editorial Manager.

Response: The ORCID iD for Lizhi Wang is :https://orcid.org/0000-0003-3074-4273, which has been authenticated in Editorial Manager.

point-by-point responses to the reviewers

Responses to Reviewer1:

Two major comments have to be given:

Reviewer1: the differences in dietary P supply were marginal , i. e., according to plasma Ca and P concentrations the dietary treatment cannot be assigned as "LP".

Response: We are so sorry, probably because our presentation is not clear enough, the reviewer did not fully understand the way of animal grouping. Please allow us to explain it here.

 In the present study, the “LP” means low true digestibility of phosphorus rather than low dietary P level. The true digestibility of dietary phosphorus was determined by a two-stage metabolic experiment according to the Difference Level Technique (DLT). The dietary levels of phosphorus in the first and second metabolic experiments were 2.2 and 3.3 g/kg, respectively. During the two metabolic experiments, all of the 28 goats were fed the same diet, and then their true digestibility of dietary phosphorus was calculated according to DLT. According to the true digestibility of dietary phosphorus, the 28 goats were grouped into HP, MP and LP group. 

Reviewer1: The whole study is not hypothesis driven. Thus, a clear concept is lacking.

Response: In fact, the hypothesis driving our study is implicit in the introduction, except that it is not clearly stated. According to your comment, to highlight the hypothesis of this study, we revised the our manuscript in L70-73 as “Therefore, in this study, we hypothesized that the digestibility of feed phosphorus in ruminants was influenced by the structure and composition of their gastrointestinal microbiota, and the 16S rRNA high-throughput sequencing technology was used to compare the bacterial diversity in the GIT of goats with different true digestibility of phosphorus (TDP).”.

Responses to Reviewer2:

Reviewer 2: L28.phyla

Response: We have revised it in L29.

Reviewer 2: L49.paying to surface water eutrophication in parts where the livestock production of monogastric is intensive [2, 3] .Phytase a type of phosphatase enzyme that catalyzes….

Response: we have revised in L50 as “contributing to surface water eutrophication in locations where there is intensive monogastric livestock production [2, 3]. Phytase is an enzyme that…”

Reviewer 2: L67.technology has not been able to complete this work. In view of this fact, in this study the 16S rRNA….

Response:we have revised in L69 as“Therefore, in this study the 16S rRNA….”

Reviewer 2: L69.true digestibility of phosphorus (TDP)….

Response:we have revised in L67 as “true digestibilities of phosphorus….’’

Reviewer 2: L71. screening to identify high-efficiency.

Response:we have revised in L71 as “screening to identify high-efficiency….”

Reviewer 2: L80. old healthy Nubian black goats with an average…. 

Response:we have revised in L80 as“old female healthy Nubian black goats with an average…”

Reviewer 2: L82.All experimental goats were kept in pens (individual) with free choice to water.

Response:we have revised in L82 as “Goats were kept in individual pens with free access to water”.

Reviewer 2: L84. …presented in (Table 1) at two times at 8:00 am and 5:00 pm.

Response:we have revised in L84 as “as shown in (Table 1) fed two times…”

Reviewer 2: L86. delete sentence starting with The formulas

Response: we have delete sentence starting with the formulas.

Reviewer 2: L90.VA: 2 937 IU, VD: 343 IU, VE: 30 IU.

Response: we have revised in L90 as“Vitamin A, Vitamin D and Vitamin E”

Reviewer 2: L97 collected. Was urine collected into acid? Or just mixed with acid after sampling?

Response: During the metabolism trial period, all the urine and feces were collected each day, and the feces and urine were sampled. The samples were mixed with hydrochloride.

Reviewer 2: L100.the 6-days metabolism

Response: we have revised in L100 as“6-day metabolism”

Reviewer 2: L103.the GIT organs (rumen, abomasum, jeujinu, cecum and colon) were unglued to collect rumen (8 tubes, each of 50 mL), abomasums (4 tubes, each 15 mL), jejunum (4 tubes, each 15 mL), cecum (4 tubes, each 15 mL) and colon (4 tubes, each 15 mL) content .

Response: we have revised inL103 as“jejunum…) were opened and the digesta was mixed and collected into tubes: rumen (8 tubes of 50 ml) ….”

Reviewer 2: L105. A tube of ruminal sample was subjected to measure the pH (PHS…

Response: we have revised in L105 as“Ruminal sample pH was measured (PHS….”

Reviewer 2: L107. Ruminal fluid was strained through four layers of cheese cloth and 20 ml were mixed with 4 mL of 25%? Meta-phosphoric acid…

110 Samples from the GIT segments were stored at -80oC for further analysis.

Response:We have revised in L117 as “Subsequently, this rumen fluid (approximately 20 mL) was mixed with 5mL solution of 25% metaphosphoric acid and preserved at -20°C for further calculation of the volatile fatty acid concentration (VFA; acetate, propionate, and butyrate).”

Reviewer 2: L110.The other remnants of GIT segment samples were stored at -80°C for further assessment. 

Response: we have revised in L110 as“Samples from the GIT segments were stored at -80℃ for further analysis.”

Reviewer 2: L116…. gas chromatography

Response: we have revised in L116 as“gas chromatography”

Reviewer 2: L119. ground to pass a 40-mesh sieve

Response: we have revised in L119 as“ground to pass a 40-mesh sieve”

Reviewer 2: L122. measure organic matter…crude ash. Neutral detergent fiber…What about mentioning specific procedure from AOAC book?

Response: we have revised in L134-135 as “Neutral detergent fiber (NDF) and acid detergent fiber (ADF) were measured according to the procedures of AOAC (method NO. 973.18 C, 1990).”

Reviewer 2: L124…..with the filter bag technique lacking of sodium sulfite and articulated with ash residual.

Response: we have revised in L134-138.

Reviewer 2: L136. True digestibility

Response: we have revised in L136 as “true digestibility”

Reviewer 2: L142. TDP of all goats were calculated with mean…

Response: we have revised in L142 as “TDP of all goats were calculated with mean…”

Reviewer 2: L142. Animals were grouped into high digestibility of phosphorus….

Response: we have revised in L142 as“Animals were grouped into high digestibility of phosphorus….”

Reviewer 2: L147. and sterile phosphate buffered saline (pH 7) was used to rinse the liquid through 4 layers of cheesecloth and collected into a sterile tube (EP). The resulting liquid was immediately centrifuged at 10,000 x g and the supernatant gently removed.

Response: we have revised in L147 as“and sterile phosphate buffered saline (pH 7) was used to rinse the liquid through 4 layers of cheesecloth and collected into a sterile tube (EP). The resulting liquid was immediately centrifuged at 10,000 x g and the supernatant gently removed.”

Reviewer 2: L152. used for DNA extraction from samples of ruminal and homogenized digesta from other organs following the manufacturers instructions using the procedure described in [19]

Response: we have revised in L152 as “used for DNA extraction from samples of ruminal and homogenized digesta from other organs following the manufacturers instructions using the procedure described in [19]”

Reviewer 2: L154. agrose electrophoresis and NanDrop 8000 spectrophotometer….

Response: we have revised in L154 as “Agarose electrophoresis and NanoDrop….”

Reviewer 2: L155….. measure DNA quality of the samples.

Response: we have revised in L155 as “measure DNA quality of the samples.”

Reviewer 2: L184…..SILVA database….

Response: we have revised in L184 as “SILVA”

Reviewer 2: L204. The TDP of the 28 goats highly varied (from 68.38% to 90.26%), and the average was 83.28% ± 6.81%.

Response: we have revised in L204 as “The TDP of the goats varied from 68.4% to 90.3% with an average of 83.3% (± 6.8%).”

Reviewer 2: L205. The TDP of the 8 animals in the HP group 89.5% ± 2.9%) was significantly higher (than for the LP group (78.4% ± 1.21%).

Response: we have revised in L229-230 as “The TDP of the 8 animals in the HP group (89.5% ± 2.9%) was significantly higher than the LP group (78.4% ± 1.21%).”

Reviewer 2: L209. was significantly higher (P<0.05) than the LP group),….

Response: we have revised in L209 as“was significantly higher (P<0.05) than the LP group”

Reviewer 2: L209….. not significantly different (P>0.05)….

Response: we have revised in L209 as “not significantly different (P>0.05)”

Reviewer 2: L230-359 While I recognize some phyla and genus, I am not familiar enough with literature to give a critical review. Yes, I have seen these in other research, but not knowledgeable enough to evaluate the research.

Response: thank you.

Reviewer 2: L368.what is the feeding standard recommendation?

Response: The recommended phosphorus level of the Chinese meat sheep and goat feeding standard (NY/T 816-2004) is 4.0-5.0 g/kg DMI. In the present study, the phosphorus dietary levels in the first and second metabolic experiments were 2.2 and 3.3 g/kg, respectively. We have revised in L394 as “…Chinese meat sheep and goat feeding standard (4.0-5.0 g/kg DMI, NY/T 816-2004)”.

Reviewer 2: L379. …. goats varied greatly ….

Response: we have revised in L379 as “goats varied greatly.”

Reviewer 2: L382…but prior to our research, there was no published report on the variation of TDP….

Response: we have revised in L382 as “but prior to our research, there was no published report on the variation of TDP…”

Reviewer 2: L383…. earlier research showed that individual variations in feed efficiency of ruminants are common [35-37].

Response: we have revised in L383 as “earlier research showed that individual variations in feed efficiency of ruminants are common […”

Reviewer 2: L393. microbial digestion is an important component of digestion in the…

Response: we have revised in L393 as “microbial digestion is an important component of digestion in the…”

Reviewer 2: L404.…. yet been reported.

Response: we have revised in L404 as “yet been reported.”

Reviewer 2: L405…. the binding of free phosphorus with some substance in feed such as protein……

Response: we have revised in L405 as “the binding of free phosphorus with some substance in feed such as protein…”

Reviewer 2: L421…nutrient availability…

Response: we have revised in L421 as “nutrient availability”

Reviewer 2: 445 CABI

Response: we have revised in L485 as “CABI”

Reviewer 2: 451 feeding

Response: we have revised in L491 as “feeding”.

Reviewer 2: 504 excretion in

Response: we have revised in L540 as “excretion in ruminants”

Reviewer 2: Why not in figure 1B and 3a CHANGE SYMBOLS. Use a solid square for HP rumen and an open square for LP rumen. Use a solid triangle for the abomasum HP and an open triangle for LP. Use a solid and open circle for jejunum and open and closed diamond for the colon.

Response: Sorry, we can't revise our article as you suggested.

---

## [Decision Letter · Decision Letter 1]

2 Mar 2020

PONE-D-19-29759R1

Relationship between True Digestibility of dietary Phosphorus and Gastrointestinal Bacteria of Goats

PLOS ONE

Dear Dr. Wang,

Thank you for submitting your manuscript to PLOS ONE. After careful consideration, we feel that it has merit but does not fully meet PLOS ONE’s publication criteria as it currently stands. Therefore, we invite you to submit a revised version of the manuscript that addresses the points raised during the review process.

Both reviewers felt that the authors had made substantial improvements to the manuscript, but were mixed on the manuscript's readiness for publication.  Reviewer 2 suggested a few minor spelling corrections, and Reviewer 1 felt that the authors had not sufficiently addressed one of their previous concerns on the interpretation.  If the authors address this comment in the manuscript, it may require substantial changes to their discussion.  If they chose not to, the authors should provide ample justification in their response as to why they felt they had already addressed this.  I encourage you to consider these additional changes, and to submit a revised manuscript.

We would appreciate receiving your revised manuscript by Apr 13 2020 11:59PM. To enhance the reproducibility of your results, we recommend that if applicable you deposit your laboratory protocols in protocols.io, where a protocol can be assigned its own identifier (DOI) such that it can be cited independently in the future. For instructions see: http://journals.plos.org/plosone/s/submission-guidelines#loc-laboratory-protocols

We look forward to receiving your revised manuscript.

Kind regards,

Suzanne L. Ishaq, PhD

Academic Editor

PLOS ONE

Journal Requirements:

In your Methods section, please provide methods of animal sacrifice and methods of anesthesia and/or analgesia.

Reviewers' comments:

Reviewer's Responses to Questions

**Comments to the Author**

1. If the authors have adequately addressed your comments raised in a previous round of review and you feel that this manuscript is now acceptable for publication, you may indicate that here to bypass the “Comments to the Author” section, enter your conflict of interest statement in the “Confidential to Editor” section, and submit your "Accept" recommendation.

Reviewer #1: (No Response)

Reviewer #2: All comments have been addressed

2. Is the manuscript technically sound, and do the data support the conclusions?

Reviewer #1: (No Response)

Reviewer #2: Yes

3. Has the statistical analysis been performed appropriately and rigorously? 

Reviewer #1: (No Response)

Reviewer #2: Yes

4. Have the authors made all data underlying the findings in their manuscript fully available?

Reviewer #1: (No Response)

Reviewer #2: Yes

5. Is the manuscript presented in an intelligible fashion and written in standard English?

Reviewer #1: (No Response)

Reviewer #2: No

6. Review Comments to the Author

Reviewer #1: Despite the authors have considered most comments originating from the first review process there still remains an open question. The authors claim some kind of interrelation between true phosphorus digestibility and the bacterial community. Based on this assumption quantitative differences in microbial protein synthesis could also be taken into account. Since the N:P ratio in microbial cell mass is rather constant differences in microbial protein synthesis would result in changes of true phosphorus digestibility. This should at least be taken into consideration in the discussion of the manuscript.

Reviewer #2: Still a few errors to correct to polish MS

99 10% sample of each samping liquid is not random

111 13}. Rumen samples were filtered through four layers of cheesecloth into 3 tubes.

120 and gas chromatography

123 ground to pass

217 were not significantly

375 as compared to pigs ( ) or chickens ( ).

7. PLOS authors have the option to publish the peer review history of their article (what does this mean?). If published, this will include your full peer review and any attached files.

Reviewer #1: No

Reviewer #2: Yes: Steve Hart

---

## [Author Response · Author response to Decision Letter 1]

16 Mar 2020

Point-by-point Responses

We greatly appreciate the insightful and careful review by the editor and reviewers and have made the following revisions.

Responses to Reviewer1

Reviewer1: Despite the authors have considered most comments originating from the first review process there still remains an open question. The authors claim some kind of interrelation between true phosphorus digestibility and the bacterial community. Based on this assumption，quantitative differences in microbial protein synthesis could also be taken into account. Since the N:P ratio in microbial cell mass is rather constant，differences in microbial protein synthesis would result in changes of true phosphorus digestibility. This should at least be taken into consideration in the discussion of the manuscript. 

Response: We have tried our best to revise our article according to the opinion of the reviewer. We agree with the reviewer that the N:P ratio in microbial cell mass is rather constant. However, the reference about “the differences in microbial protein synthesis would result in changes of true phosphorus digestibility” is not much. At last, we revise our manuscript in L432-440 as “Besides being relate to the ability of rumen microorganisms to produce phytase, the correlation between gastrointestinal bacteria and TDP in goats found in this study may also be related to the difference in rumen microbial protein synthesis between groups. Previous research found that the ratio of nitrogen to phosphorus in ruminal microbial cell mass is rather constant [49]，differences in microbial protein synthesis would result in changes of true phosphorus digestibility. Unfortunately, this study did not measure the activity of microbial phytase in the GIT and the amount of microbial protein synthesis in the rumen. Therefore, it was uncertain that the improvement of feed phosphorus digestibility was achieved through the increases of microbial phytase secretion or microbial protein synthesis.”

Responses to Reviewer2:

Reviewer 2: 99 10% sample of each samping liquid is not random

Response: According to your comment, we revise our manuscript in L98-100 as “During the metabolism trial period all the urine and feces were collected. For nitrogen fixation, the 10% of each urine and feces samples were collected each day and then mixed with 10ml of hydrochloride (10%, v/v).”

Reviewer 2: 111 13}. Rumen samples were filtered through four layers of cheesecloth into 3 tubes.

Response: According to your comment, we revise our manuscript in L112-113 as “Rumen samples were filtered through four layers of cheesecloth into 3 tubes.”

Reviewer 2: 120 and gas chromatography

Response: According to your comment, we revise our manuscript in L120 as “and gas chromatography”.

Reviewer 2: 123 ground to pass

Response: According to your comment, we revise our manuscript in L 123 as “ground to pass”.

Reviewer 2: 217 were not significantly

Response: According to your comment, we revise our manuscript in L217 as “were not significantly”. 

Reviewer 2:375 as compared to pigs ( ) or chickens ( ).

Response: According to your comment, we revise our manuscript in L 375 as “which was higher as compared to pigs [31] or chickens [32].”.

---

## [Decision Letter · Decision Letter 2]

27 Mar 2020

PONE-D-19-29759R2

Relationship between True Digestibility of dietary Phosphorus and Gastrointestinal Bacteria of Goats

PLOS ONE

Dear Dr. Wang,

Thank you for submitting your manuscript to PLOS ONE. After careful consideration, we feel that it has merit but does not fully meet PLOS ONE’s publication criteria as it currently stands. Therefore, we invite you to submit a revised version of the manuscript that addresses the points raised during the review process.

I would to thank the authors for their attentiveness in addressing reviewer comments.  Both reviewers are satisfied with the changes that have been made, and I agree that the manuscript is acceptable for publication.  One reviewer noted several grammatical things to correct, and these were numerous enough that I felt it would be easier to correct these as a new submission rather than during the proofing process.  Please make the recommended grammatical changes, and I will accept the resubmission without sending it for additional review.

In addition, to comply with PLOS ONE's policy on animal research (https://journals.plos.org/plosone/s/submission-guidelines#loc-animal-research), please provide methods of animal sacrifice and/or methods of anesthesia and/or analgesia in your Methods section.

We would appreciate receiving your revised manuscript by May 10 2020 11:59PM. To enhance the reproducibility of your results, we recommend that if applicable you deposit your laboratory protocols in protocols.io, where a protocol can be assigned its own identifier (DOI) such that it can be cited independently in the future. For instructions see: http://journals.plos.org/plosone/s/submission-guidelines#loc-laboratory-protocols

We look forward to receiving your revised manuscript.

Kind regards,

Suzanne L. Ishaq, PhD

Academic Editor

PLOS ONE

Reviewers' comments:

Reviewer's Responses to Questions

**Comments to the Author**

1. If the authors have adequately addressed your comments raised in a previous round of review and you feel that this manuscript is now acceptable for publication, you may indicate that here to bypass the “Comments to the Author” section, enter your conflict of interest statement in the “Confidential to Editor” section, and submit your "Accept" recommendation.

Reviewer #1: All comments have been addressed

Reviewer #2: All comments have been addressed

2. Is the manuscript technically sound, and do the data support the conclusions?

Reviewer #1: Yes

Reviewer #2: Yes

3. Has the statistical analysis been performed appropriately and rigorously? 

Reviewer #1: Yes

Reviewer #2: Yes

4. Have the authors made all data underlying the findings in their manuscript fully available?

Reviewer #1: Yes

Reviewer #2: Yes

5. Is the manuscript presented in an intelligible fashion and written in standard English?

Reviewer #1: Yes

Reviewer #2: No

6. Review Comments to the Author

Reviewer #1: All comments have been adressed adequately and therefore, this manuscript is now in a version to contribute to new findings regarding P homeostasis in ruminants.

Reviewer #2: There are a few changes to polish the MS.

Line correction

21 Relationship instead of connection

26 Delete And aBacterial 16S rNA gene amplicons were sequenced from the rumen, abomasum, jejunum, cecum and Colten contents of the goats.

66 accomplish this instead of complete this

94 stages

141 true

233 A total of 5,711,204 high quality sequences were produced by the 16S rNA analysis, 915,084 in the rumen, 931,464 in the abomasum, 900,688 in the jejunum, 909,122 in the colon and 813,476 in the cecum.

236 in the rumen, the abomasum, the jejunum etc.

286 top 10 most plentiful

324 significantly

325 significantly compared

326 significantly

330 significantly

331 compared

337 in relative

339 significantly significantly

341 significantly

342 as compared

342 significantly

433 The correlation between TDP and microbial genus may not only be caused by the ability of the microbial genus to produce phytase, but may also be related to the difference….

7. PLOS authors have the option to publish the peer review history of their article (what does this mean?). If published, this will include your full peer review and any attached files.

Reviewer #1: No

Reviewer #2: Yes: Steve Hart

---

## [Author Response · Author response to Decision Letter 2]

28 Mar 2020

Point-by-point Responses

We greatly appreciate the insightful and careful review by the editor and reviewers and have made the following revisions.

Responses to editor:

Editor: In addition, to comply with PLOS ONE's policy on animal research (https://journals.plos.org/plosone/s/submission-guidelines#loc-animal-research), please provide methods of animal sacrifice and/or methods of anesthesia and/or analgesia in your Methods section.

Response: We revised our MS as requested in L75-78 Materials and methods section as “The research procedure (Protocol Number: 2018-12825-01) used in current study was approved by Animals policy and welfare committee of Agricultural research organization of Sichuan province China and in agreement with rules of the Animal Care and Ethical Committee of the Sichuan Agricultural University (SAU).”, and in L108-110 as “Then the goats were slaughtered. Pre-harvest handling was conducted in accordance with good animal welfare practices, and slaughtering procedures followed actual Chinese law on animal production and sanitary inspection.”

Responses to Reviewer2:

Reviewer #2: There are a few changes to polish the MS.

Reviewer 2: 21 Relationship instead of connection

Response: According to your comment, we revise our manuscript in L21 as “relationship….”

Reviewer 2: 26 Delete And aBacterial 16S rNA gene amplicons were sequenced from the rumen, abomasum, jejunum, cecum and Colten contents of the goats

Response: According to your comment, we revise our manuscript in L26 as “The bacterial 16S rRNA gene amplicons were sequenced from the rumen, abomasum, jejunum, cecum and colon contents of the goats.”

Reviewer 2: 66 accomplish this instead of complete this

Response: According to your comment, we revise our manuscript in L66 as “accomplish …”

Reviewer 2: 94 stages

Response: According to your comment, we revise our manuscript in L94 as “ stages…”

Reviewer 2: 141 true

Response: According to your comment, we revise our manuscript in L141 as “ true….”

Reviewer 2: 233 A total of 5,711,204 high quality sequences were produced by the 16S rNA analysis, 915,084 in the rumen, 931,464 in the abomasum, 900,688 in the jejunum, 909,122 in the colon and 813,476 in the cecum.

Response: According to your comment, we revise our manuscript in L233 as “A total of 5,711,204 high quality sequences were produced by the 16S rNA analysis, 915,084 in the rumen, 931,464 in the abomasum, 900,688 in the jejunum, 909,122 in the colon and 813,476 in the cecum.” 

Reviewer 2: 236 in the rumen, the abomasum, the jejunum etc.

Response: According to your comment, we revise our manuscript in L236 as “were recognized in the rumen, in the abomasum 5,810, in the jejunum 5,611, in the colon 8,886 and in the cecum……”

Reviewer 2: 286 top 10 most plentiful

Response: According to your comment, we revise our manuscript in L286 as “top 10 most plentiful…..”

Reviewer 2: 324 significantly

Response: According to your comment, we revise our manuscript. 

Reviewer 2: 325 significantly compared

Response: According to your comment, we revise our manuscript in L325 as “was significantly (p<0.05) lower in the LP compared to HP group……”

Reviewer 2: 326 significantly

Response: According to your comment, we revise our manuscript in L326.

Reviewer 2: 330 significantly

Response: According to your comment, we revise our manuscript in L330.

Reviewer 2: 331 compared

Response: According to your comment, we revise our manuscript in L331.

Reviewer 2: 337 in relative

Response: According to your comment, we revise our manuscript in L337 as “in relative..”

Reviewer 2: 339 significantly significantly

Response: According to your comment, we revise our manuscript in L339.

Reviewer 2: 341 significantly

Response: According to your comment, we revise our manuscript in L341.

Reviewer 2: 342 as compared

Response: According to your comment, we revise our manuscript in L342.

Reviewer 2: 342 significantly

Response: According to your comment, we revise our manuscript in L342.

Reviewer 2: 433 The correlation between TDP and microbial genus may not only be caused by the ability of the microbial genus to produce phytase, but may also be related to the difference….

Response: According to your comment, we revise our manuscript in L433 as “The correlation between TDP and microbial genus may not only be caused by the ability of the microbial genus to produce phytase, but may also be related to the difference….”

---

## [Editor Report · Decision Letter 3]

29 Apr 2020

PONE-D-19-29759R3

Relationship between True Digestibility of dietary Phosphorus and Gastrointestinal Bacteria of Goats

PLOS ONE

Dear Dr. Wang,

Thank you for submitting your manuscript to PLOS ONE. After careful consideration, we feel that it has merit but does not fully meet PLOS ONE’s publication criteria as it currently stands. Therefore, we invite you to submit a revised version of the manuscript that clarifies the following information:

-how and where the goats were slaughtered

-the method of anaesthesia or analgesia used, if relevant

Please address these in the methods section, as these revisions were previously requested and are required to move forward with publication.

We would appreciate receiving your revised manuscript by Jun 13 2020 11:59PM. To enhance the reproducibility of your results, we recommend that if applicable you deposit your laboratory protocols in protocols.io, where a protocol can be assigned its own identifier (DOI) such that it can be cited independently in the future. For instructions see: http://journals.plos.org/plosone/s/submission-guidelines#loc-laboratory-protocols

We look forward to receiving your revised manuscript.

Kind regards,

Suzanne L. Ishaq, PhD

Academic Editor

PLOS ONE

---

## [Author Response · Author response to Decision Letter 3]

3 May 2020

Comments: -how and where the goats were slaughtered

-the method of anaesthesia or analgesia used, if relevant

Response: We have revised as per suggestion thanks.

The goats were slaughtered at slaughter house in Teaching and Research Base, Animal Nutrition Institute, Sichuan Agricultural University. All the goats were slaughter according to the American Veterinary Medical Association (AVMA) guidelines for the humane slaughter of animals.

Note: (The animals were stunned by captive bolt and the exsanguination from the jugular vein was carried out)

---

## [Editor Report · Decision Letter 4]

6 May 2020

Relationship between True Digestibility of dietary Phosphorus and Gastrointestinal Bacteria of Goats

PONE-D-19-29759R4

Dear Dr. Wang,

We are pleased to inform you that your manuscript has been judged scientifically suitable for publication and will be formally accepted for publication once it complies with all outstanding technical requirements. Thank you for supplying that additional detail to the methods section.

With kind regards,

Suzanne L. Ishaq, PhD

Academic Editor

PLOS ONE
---

## [Editor Report · Acceptance letter]

8 May 2020

PONE-D-19-29759R4 

Relationship between True Digestibility of dietary Phosphorus and Gastrointestinal Bacteria of Goats 

Dear Dr. Wang:

I am pleased to inform you that your manuscript has been deemed suitable for publication in PLOS ONE. Congratulations! Your manuscript is now with our production department. 

With kind regards,

on behalf of

Dr. Suzanne L. Ishaq 

Academic Editor

PLOS ONE